# Reporting health services research to a broader public: An exploration of inconsistencies and reporting inadequacies in societal publications

**Reinie G. Gerrits** (ID), **Michael J. van den Berg, Anton E. Kunst, Niek S. Klazinga, Dionne S. Kringos** (ID)*

Department of Public and Occupational Health, Amsterdam UMC, University of Amsterdam, Amsterdam Public Health Research Institute, Amsterdam, The Netherlands

* d.s.kringos@amsterdammumc.nl

## Abstract

### Introduction

Little is known about the accuracy of societal publications (e.g. press releases, internet postings or professional journals) that are based on scientific work. This study investigates a) inconsistencies between scientific peer-reviewed health services research (HSR) publications and non-scientific societal publications and b) replication of reporting inadequacies from these scientific publications to corresponding societal publications.

### Methods

A sample of HSR publications was drawn from 116 publications authored in 2016 by thirteen Dutch HSR institutions. Societal publications corresponding to scientific publications were identified through a systematic internet search. We conducted a qualitative, directed content analysis on societal publications derived from the scientific publications to assess both reporting inadequacies and determine inconsistencies. Descriptive frequencies were calculated for all variables. Odds ratios were used to investigate whether inconsistencies in societal publications were less likely when the first scientific author was involved.

### Results

We identified 43 scientific and 156 societal publications. 94 societal publications (60.3%), (associated with 32 scientific publications (74.4%)) contained messages that were inconsistent with the scientific work. We found reporting inadequacies in 22 scientific publications (51.2%). In 45 societal publications (28.9%), we found replications of these reporting inadequacies. The likelihood of inconsistencies between scientific and societal publications did not differ when the latter explicitly involved the first scientific author, (OR = 1.44, CI: 0.76–2.74); were published on the institute's or funder's website, (OR = 1.32, CI: 0.57–3.06); published with no involvement of a scientific author, (OR = 0.52, CI: 0.25–1.07).

**Data Availability Statement:** All relevant data are available within the project files in the Figshare

public repository: https://doi.org/10.21942/uva.9255335.

**Funding:** This study was funded by grant number 445001003 from the Netherlands Organisation for Health Research and Development (ZonMw). The funder had no role in the study design, the collection, analysis and interpretation of data, the writing of the manuscript or the decision to submit the paper for publication. All authors had full access to the data during the conduct of the study and they take responsibility for the integrity of the data and the analysis.

**Competing interests:** The authors have declared that no competing interests exist.

## Conclusion

To improve societal publications, one should examine both the consistency with scientific research publications and ways to prevent replication of scientific reporting inadequacies. HSR institutions, funders, and scientific and societal publication platforms should invest in a supportive publication culture to further incentivise the responsible and skilled involvement of researchers in writing both scientific and societal publications.

## Background

In academia, scientific research publications are an important source of knowledge, as well as a means of research dissemination [1]. Outside the research community, however, most people take note of research findings through non-scientific, societal publications such as press releases, newspapers, social media, internet postings or professional journals [2–4]. The content of societal publications impacts the thinking, debates and decisions of the general public, as well as those of patients, health professionals and policymakers [4–6]. Consequently, researchers who publish a scientific paper are increasingly incentivised to translate their findings into a corresponding societal publication, in order to reach broader, often non-academic audiences [7].

By necessity, the authors of societal publications simplify scientific messages and conclusions for their lay target group [8]. Although this can be done in a responsible manner, it does present a risk for misrepresentation and misinterpretation of the research findings [9]. Previous studies on biomedical publications concluded that unjustified causal claims are introduced in 20% to 33% of press releases, and that 40% of news articles give more explicit health advice to the readers than was expressed in the underlying scientific publication [10–14].

In health services research (HSR), less is known about the potential misrepresentation or misinterpretation of evidence in societal publications. HSR aims to provide usable evidence for policy and for management of health and health care [5]. This practice-oriented ambition amplifies the importance of accuracy in all messages and conclusions relayed in societal publications [15].

Researchers are often expected to have a societal impact beyond their scientific impact. Funders of HSR increasingly demand strategies to achieve a societal impact. Methods for measuring impact are being developed and refined [16–18]. Researchers, however, may lack the experience or capability to write responsible societal publications that accurately reflect their scientific findings [10, 19]. Some previous research has concluded that a researcher's involvement is not associated with better societal publications [10]. Researchers may have difficulty working with journalists, or they may lack the ability to explain their findings in simple terms [19]. Moreover, fellow researchers may take a critical view of colleagues who invest considerable time in media attention, and thus discourage them to put significant efforts in writing societal publications [20].

Messages and conclusions are too often poorly reported in the scientific publications [21, 22]. In a previous assessment of peer-reviewed HSR publications written in the Netherlands for an international academic audience, we found per publication a median of 6 out of 35 possible 'questionable research practices' in the reporting of messages and conclusions [23]. In the current study, these questionable research practices will be called 'reporting inadequacies'. They include conclusions that are insufficiently supported by the research results, recommendations that are not justified and limitations that are inadequately explained [23]. Even if a

researcher tries to avoid inconsistencies in a subsequent societal publication, such reporting inadequacies in the original work may well find their way to a broader audience. As the scientific publication is used as the standard, reporting inadequacies will likely be copied or 'replicated' to societal publications.

Given the potential impact of societal publications on policy and practice, knowledge of responsible reporting in societal publications, and how researchers can achieve it, is important for the HSR community [24]. Such knowledge is currently inadequate [10, 14]. Whereas the previous studies in the field of biomedicine focused largely on press releases and newspapers, broader insights are needed into the full scope of societal HSR publications, including information sources such as fact sheets, web pages and articles in professional journals.

The aims of this study are to explore

1. whether societal publications on HSR are consistent with the messages reported in the underlying research papers

2. whether reporting inadequacies in scientific HSR publications are replicated in societal publications

3. whether fewer inconsistencies occur in societal publications if the first scientific author is involved in writing them.

## Methods

In a collaboration funded by the Netherlands Organisation for Health Research and Development (ZonMw), thirteen Dutch academic and non-academic HSR institutions (see Acknowledgements section for the listing) took part in several studies designed to promote responsible reporting. The present study builds on the results of a previous study that identified reporting inadequacies in scientific publications [23].

No patients or human participants were involved in this study. A waiver for ethical approval was obtained for this study from the Medical Ethics Review Committee at Amsterdam UMC.

To investigate inadequacies in research reporting and inconsistencies between scientific and societal publications, we employed a mixed-methods approach. We first conducted a directed qualitative content analysis of scientific HSR publications and related societal publications that derived from them, followed by a quantitative description of the results.

### Subsample of scientific publications

We based our selection of scientific publications on a random sample of 116 such publications authored in 2016 by researchers from the thirteen participating HSR institutions.

In short, complete publications lists were obtained from all institutions, from which a total of 717 scientific HSR publications were identified, applying commonly used definitions of HSR from Plochg and colleagues [25] and Lohr and Steinwachs [26]. Two researchers independently selected publications based on these definitions. Publications that were selected by both researchers were included. Remaining publications were included or excluded after review by the full research group. A sample of 116 publications was included. Two researchers independently assessed those publications for inadequacies in the reporting of messages and conclusions, using a validated checklist of 35 possible inadequacies. Each inadequacy was recorded on an assessment form. During periodic consensus meetings, the reviewers compared their assessment of all items. Inconsistencies between the individually assessed reporting inadequacies were identified, discussed and adapted. Any remaining disagreements (n = 2) were resolved by a third, senior, researcher. The list of assessed reporting inadequacies is

added to S1 Appendix. An extensive description of the sampling, development of the assessment tool and the assessment of these publications has been published elsewhere [23].

The current study confined itself to scientific writings that had one or more associated societal publications. Quantitative, qualitative and mixed-methods studies were included. We aimed to include publications with the most and the least reporting inadequacies, to allow for detection of differences in the replication reporting inadequacies. We sampled scientific publications with relatively high and low numbers of reporting inadequacies based on the median of inadequacies per publication ('high' being more than 6 and 'low' fewer than 6 reporting inadequacies). We did not include scientific publications with the median of 6 reporting inadequacies.

## Sample of societal publications

Societal publications corresponding to scientific publications were identified through a systematic internet search. We included societal publications that (1) were in the public domain, (2) contained messages on the same research as the corresponding scientific publication (including statements on the results, conclusions, discussion, recommendations or implications) and (3) were written in Dutch (aside from social media messages communicating the title of the publication). Societal publications corresponding to the specific scientific publication were identified based on the content of the publication, containing mention of the author, research program, specific study title or aim.

For each scientific publication, a variety of internet sources were consulted, following a systematic search strategy. We searched or consulted (1) specific institute websites, funders' websites and Altmetrics; (2) document databases of Dutch government and parliament (including www.rijksoverheid.nl, www.tweedekamer.nl, and https://zoek.officielebekendmakingen.nl/); (3) databases of Dutch popular science periodicals (https://www.skipr.nl/zoeken?q, https://www.medischcontact.nl) and a periodical aimed at medical professionals https://www.ntvg.nl/zoeken); (4) an existing database of Dutch newspaper articles (www.lexisnexis.nl); (5) public social media platforms (LinkedIn and Twitter) of the authors and the institutes; (6) the Google search engine, to identify publications from further sources. In the Google search, we entered search terms (see next section) and followed all links provided in the first 30 results, as we did not expect to find relevant societal publications beyond that ranking. To ensure that earlier searches did not affect the Google search, our browser history data, including cookies, were deleted beforehand. All internet sources were accessed in the month of August 2018.

## Search terms and filters

For each scientific publication, specific search terms were derived from Dutch translations of key terms in the title and abstract; also included were the name of the first author's institution, the authors' names and the funder(s). Any new key terms found during the search were added.

Search strings were used if the database enabled the use of logical operators. Because results of a study may be reported prior to the appearance of the scientific publication, we included societal publications appearing up to two years beforehand and one year afterwards (presuming that all societal publications would appear within a year). All search terms were discussed and approved by two members of the project team (RG and NK).

## Analyses of messages in societal publications

Societal publications were analysed using a directed content analysis approach [27]. A directed content analyses is a more structured approach than traditional content analyses. It starts with an initial coding scheme based upon prior knowledge [27]. First, we identified distinct

messages and conclusions in the societal publications that related to the corresponding scientific publication. Messages could be a single sentence or a section of the text elaborating on the same topic; a single research result or a concluding statement was marked as a distinct message. Multiple messages might be identified in a single societal publication.

Second, we assessed whether the message in the societal publication was consistent with that in the corresponding scientific work. A message was considered consistent if it conveyed the same meaning as the scientific results, discussion or conclusion and if no changes, additions or subtractions had been made with respect to the content of the scientific assertion. In line with our directed content analyses approach, an initial coding scheme based on the possible inconsistencies was prepared, informed by other checklists for public reporting [12, 28]. To discover other types of inconsistencies not included in those checklists, we iteratively improved the coding scheme during the first stages of coding, adding new aspects that emerged during the coding.

Third, we determined whether a message in the societal publication replicated a reporting inadequacy in the scientific publication. Messages in the societal writings were compared to any reporting inadequacies recorded during the previous assessment of the corresponding scientific paper [23]. A message that identically reproduced the reporting inadequacy was marked as a 'replicated reporting inadequacy'.

Finally, for each societal publication we gauged the likelihood of the first scientific author's involvement ('named as author', 'published on institute or funder web page' or 'no involvement').

One coder (RG) performed the analysis. The identification of the messages and the coding method of the first ten publications were checked by project members (NK, DK and MB) and thoroughly discussed until the coding method and scheme had been agreed. To ensure consistency of analyses, we had 10 per cent ($n = 16$) of the analysed societal publications randomly checked by DK. Prompted by the check, we decided to revisit the final 15 societal publications to improve possible inconsistencies with earlier codes, and to correct one identified replicated reporting inadequacy. Analyses were conducted in MAXQDA.

All messages were coded as categorical variables. Counts of inconsistencies and reporting inadequacies were calculated within the program MAXQDA. S2 Appendix provides the coding scheme.

### Statistical analyses

The codes in MaxQDA were transferred to a SPSS dataset. Odds ratios were calculated to compare the frequencies of inconsistencies in societal publications (1) authored by the first scientific author, (2) published on the institute's or funder's web page or (3) published with no involvement of a scientific author who was part of the scientific publication. Three tests were performed, each comparing one category to the other two categories combined. A societal publication was deemed 'inconsistent' if at least one message in it was identified as inconsistent.

## Results

### Characteristics of analysed publications

We conducted the structured internet search until we identified 46 scientific publications (23 with high and 23 with low inadequacies) that had associated societal publications. We identified the included 46 publications after examining 84 scientific publications in our sample (46 with the highest number of inadequacies and 38 with the lowest number of inadequacies). We examined 188 societal publications obtained in our internet search and excluded 32 of them from further analysis because they only described the applied methodology without reference to study results. That left three further scientific publications without corresponding societal

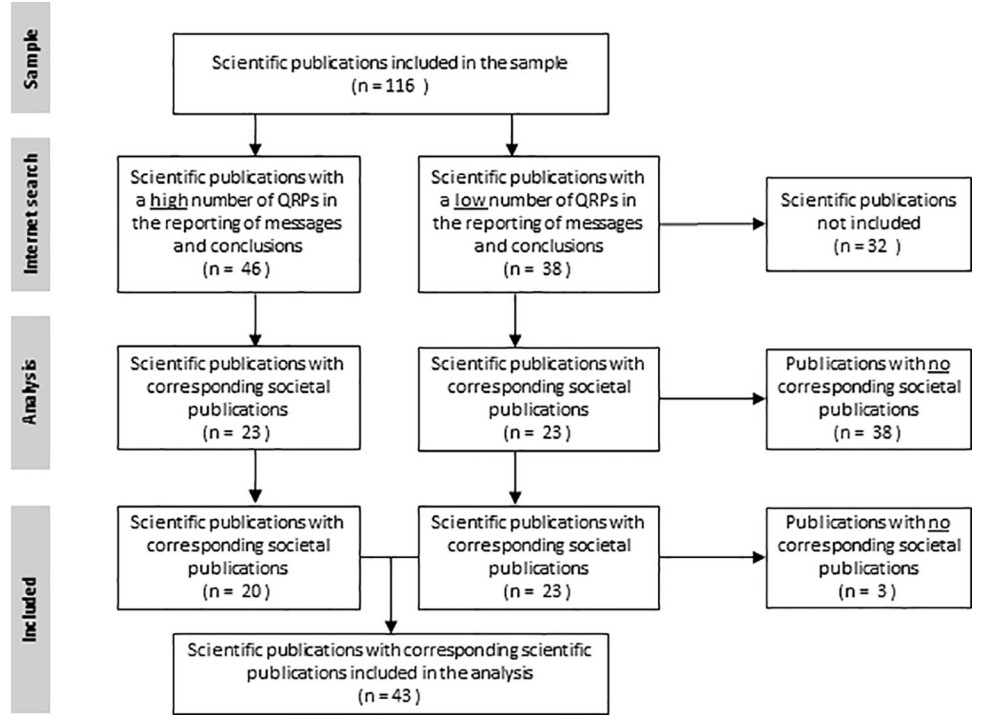

**Fig 1. Flow chart inclusion and exclusion process.**

ones, so that 43 scientific publications and 156 societal publications remained in the final sample. Fig 1 provides a flow diagram describing the inclusion and exclusion process.

Table 1 shows the characteristics of the included societal publications.

**Table 1. Characteristics of the analysed societal publications.**

| Type of societal publication | Societal publications, *n* (%) |
|---|---|
| News message | 37 (23.7) |
| Institute web page | 21 (13.5) |
| Magazine | 20 (12.8) |
| Social media | 19 (12.2) |
| Professional journal | 19 (12.2) |
| Report | 13 (8.3) |
| Thesis summary | 9 (5.8) |
| Funder web page | 8 (5.1) |
| Fact sheet | 7 (4.5) |
| Video | 2 (1.3) |
| PowerPoint slides | 1 (0.6) |
| **Linkage to scientific reporting inadequacies** | **Societal publications, *n* (%)** |
| Linked to high ($>6$) reporting inadequacies in scientific publications ($n = 20$) | 62 (39.7) |
| Linked to low ($<6$) reporting inadequacies in scientific publications ($n = 23$) | 94 (60.3) |
| **Publication time of the societal publication** | |
| Before publication of scientific publication | 30 (19.2) |
| After publication of scientific publication | 126 (80.8) |
| **Total societal publications** | **156 (100.0)** |

Scientific publications had a maximum of 14 associated societal publications, a minimum of 1 and a median of 3. The majority (*n* = 27) had 1 to 3 societal publications, 11 had 4 to 6 societal publications, and 5 had more than 6. A total of 60.3% of the societal publications corresponded to a scientific publication with low reporting inadequacies; 39.7% were linked to a publication with high inadequacies.

## Consistency of reporting between societal and scientific publications

In the 156 analysed societal publications, we identified 577 distinct messages, 342 (59.3%) of which were consistent with the corresponding message in the underlying scientific publication. Multiple types of inconsistencies were sometimes identified within a single societal publication, such as altered conclusions or differing interpretations of outcomes.

**Inconsistencies in conclusions.** We identified inconsistencies in conclusions in 64 out of 156 societal publications (41.0%). The majority of inconsistencies in societal publications concerned the scientific research conclusions. Conclusions might be altered entirely (in 13 publications) or partially, where some words or sentences were added (in 14) (e.g. a study concluding that patients have trouble speaking up, concludes patients 'never' speak up.) Moreover, conclusions were presented that were not underpinned by the scientific results or discussion. Some publications gave interpretations of the study results that were not included in the scientific work (17 publications) or added strong rhetoric to conclusions that was inconsistent with the scientific verdict (e.g. implying that a problem was worse; 12 publications). Some societal publications put forward conclusions that could not have resulted from the scientific study (21 publications) or that were derived from the introduction (4 publications) (e.g. the societal publication states that the study concludes smoking is harmful, while the scientific publication concludes on measures that would reduce harm). Some societal publications even contradicted the conclusions from the corresponding scientific publications (10 societal publications) (e.g. the societal publication states improvement in attitudes, while the scientific publication states attitudes showed no change).

**Inconsistencies in results.** We identified inconsistences in 38 out of 156 societal publications (24.4%). In various societal publications, new results were introduced that were not reported in the corresponding scientific publication (22 societal publications) (e.g. mention of additional secondary analyses not included in the scientific publication, or mention of an additional theme not included in a qualitative analyses). Results were reported in different combinations that changed the interpretation of the results (in 7 publications) (e.g. from a survey study the societal publication picks a different combination of results than the main results mentioned in the conclusion of the scientific publications). Some quantitative results were altered with respect to the figures given in the scientific publication (e.g. differing exact percentages) or qualitative results were worded differently, causing the core meaning of the scientific findings to change (13 publications). Non-significant results were presented as significant (in 1 publication), such as referring to a 'lesser effect from this intervention', whereas no effect had been indicated or argued in the scientific publication.

**Inconsistencies in recommendations.** We identified inconsistencies in recommendations in 25 out of 156 societal publications (16.0%). Recommendations differed from those made in scientific publications in three ways: (1) entirely new recommendations for policy or practice were put forward in the societal publication, whilst not mentioned in the scientific publication (in 21 societal publications) (e.g. the scientific publication mentions no concrete recommendation, but the societal publication concretely recommends the implementation of a national policy); (2) relevant limitations of the recommendations given in the scientific publication were omitted in the societal publication (1 publication) (e.g. the societal publications

recommends implementation of the program. However, the scientific publication mentions this is only effective if another party takes action, which was considered unlikely); (3) elements of recommendations given in societal publications were omitted in the scientific publication (6 societal publications) (e.g. the societal publication states what the results cannot be used for, while the scientific publication does not).

**Inconsistencies in the reporting of conditions in the conclusion.** We identified inconsistencies in the reporting of conditions in the conclusion in 4 out of 156 societal publications (2.6%). In these four societal publications, conditions affecting the study conclusions were left out, although the scientific publication explicitly made the conclusions subject to those conditions (e.g. with an 'if' or 'when' statement as part of the conclusion).

**Inconsistencies in the reporting of implications for policy and practice.** Implications for policy and practice must be differentiated from recommendations: implications describe the importance of the findings for policy and practice, while recommendations are specific measures that could improve policy and practice. In four societal publications out of 156 societal publications (2.6%), implications for policy and practice were reported that were not mentioned in the scientific publication (e.g. the societal publication mentions the possible implication that the findings might aid in saving time in healthcare provision, although this was not mentioned in the scientific publication).

**Inconsistencies in the reporting of causality.** Four societal publications out of 156 publications (2.6%) contained statements on potential causal relationships that were not mentioned in the scientific publication, and causality was implied without mention of mediating influences (e.g. one societal publication mentioned a correlation, as opposed to the scientific publication, where a causal relationship was claimed).

**Inconsistencies in reporting generalisations.** Three societal publications out of 156 (1.7%) generalised findings beyond the setting described in the scientific publication–to a different time period or geographical location, as from an urban to a rural setting; to different population characteristics such as gender, ethnicity or age; or to settings or institutions not included in the research.

**Objectives not included in the scientific publication.** In one societal publication out of 156 societal publications (0.6%), a study objective was added and discussed that was not included in the scientific publication (nor in any related research project).

## Replication of reporting inadequacies from scientific to societal publications

Reporting inadequacies found in 22 of the 43 (51.2%) included scientific publications were reproduced in corresponding societal publications. From our checklists of inadequacies in scientific reporting, we identified nine types of inadequacies that were replicated in societal publications:

- 'Conclusions do not adequately reflect the findings as presented in the results section' (from 23.3%, $n = 10$ of the 43 scientific publications)

- 'Recommendations do not adequately reflect the results in the context of the referenced literature' (26.3%, $n = 7$ of the 43 scientific publications)

- 'The title does not adequately reflect the main findings' (9.3%, $n = 4$); that is, the inadequate title of the scientific publication was replicated in a societal publication.

- 'The sampling methodology does not allow the type of generalisation provided' (7%, $n = 3$ of the 43 scientific publications)

- 'The conclusions in the abstract do not adequately reflect the conclusions in the main text [of the scientific publication]' (4.7%, $n = 2$ of the 43 scientific publications); that

is, inadequately reported conclusions from the abstract were replicated in a societal publication.

- 'A potential causal relationship claimed in the discussion paragraph is not justified' (4.7%, $n = 2$ of the 43 scientific publications)

- 'Implications for policy and practice do not adequately reflect the results in the context of the referenced literature' (2.3%, $n = 1$ of the 43 scientific publications)

- 'The abstract does not adequately reflect the main findings' (2.3%, $n = 1$ of the 43 scientific publications)

- 'Generalising findings to geographical locations not included in the original study is not justified' (2.3%, $n = 1$ of the 43 scientific publications)

### The role of the first scientific author in inconsistencies appearing in societal publications

From our sample of 43 scientific publications, 26 first authors were named as authors of a societal publication (60.5%). Some 34 scientific publications were linked to at least one societal publication that did not explicitly state involvement of the first author (79.1%). Research from 20 scientific publications was summarised on the website of a research institute or funder without explicit mention of the involvement of the author (46.5%).

Odds ratios were calculated comparing the frequencies of inconsistencies in societal publications (1) authored by the first author of the scientific publication, (2) published on the institute's or funder's website, and (3) published elsewhere without explicit involvement of the scientific author. No associations were found between the number of inconsistencies in societal publications and any of those three conditions (Table 2).

### Consistencies and replicated reporting inadequacies across scientific publications

Following our analyses, the sample of scientific publications ($N = 43$) could be broken down into four unique groups in relation to the associated societal publications:

**Table 2. The role of the first scientific author in the occurrence of inconsistencies between a societal and a scientific publication; odds ratio for each category (row) compared to both others ($N = 156$).**

| Type of societal publication | Author involvement, n (%) | At least one inconsistency found, N (%) | | Odss ratios and confidence intervals |
|---|---|---|---|---|
| | | Yes | No | |
| No involvement of scientific author | 84 (53.9) | 54 (64) | 30 (36) | OR[a] 1.44 |
| | | | | CI[b] 0.76–2.74 |
| Authored by first scientific author | 43 (27.6) | 21 (49) | 22 (51) | OR[a] 0.52 |
| | | | | CI[b] 0.25–1.07 |
| Published on institute or funder website | 29 (18.6) | 19 (66) | 10 (34) | OR[a] 1.32 |
| | | | | CI[b] 0.57–3.06 |
| Total | 156 (100) | 94 (60) | 62 (40) | |

[a] Odds Ratio

[b] Confidence Interval.

- All corresponding societal publications were consistent and did not replicate any reporting inadequacies ($n = 7$ scientific publications associated with 45 societal publications).

- Corresponding societal publications replicated reporting inadequacies, but were fully consistent with the scientific publication ($n = 4$ scientific publications associated with 17 societal publications).

- Corresponding societal publications were inconsistent, but did not replicate reporting inadequacies ($n = 15$ scientific publications associated with 67 societal publications).

- Corresponding societal publications were inconsistent with the scientific publication and replicated reporting inadequacies ($n = 17$ scientific publications associated with 27 societal publications).

## Discussion

The aims of this study were to explore (1) whether societal publications on health services research are consistent with the messages communicated in the original scientific research paper, (2) whether apparent reporting inadequacies in scientific HSR publications are replicated in societal publications, and (3) whether fewer inconsistencies occur in societal publications if they are authored by the first author of the scientific work. 60.3% of the 156 societal publications (associated with 74.4% of the scientific publications) contained messages that were inconsistent with the scientific work. Reporting inadequacies in 51.2% ($n = 22$) of the scientific publications were replicated in corresponding societal publications ($n = 45$, 28.9%). The involvement of the first author was not associated with more consistent societal publications.

Our results indicate that, as previously shown for biomedicine, the field of HSR faces issues with (mis)representation and (mis)interpretation of the research findings, as reported in societal publications [10–14]. Such issues arise not only in news articles or press releases, but also in societal publications such as professional journal articles aimed directly at policy and practice.

### Limitations

As our coding scheme was not specifically designed to identify causality, we have likely underestimated the occurrence of causal claims. The coding schemes used in previous studies, though very extensive, would not have been adequate for detecting many types of inconsistencies, such as rhetorical formulations of conclusions or diverging interpretations of results, as we have done in this study. In addition, the existing coding schemes would not have been suitable for HSR, as different types of systematic research were addressed here, including qualitative and mixed methods studies, and different types of societal publications were included in our analyses, such as tweets and fact sheets.

We analysed whether assertions in a societal publication were consistent with those in the corresponding scientific publication. We did not assess omitted messages; that is, we did not identify scientific reporting inadequacies attributable to the absence of common elements such as limitations, recommendations or contradictory evidence. Consequently, we also did not take a normative stand on whether those items should have been included in a societal publication. Such would not have been feasible considering the variety of societal publications studied, ranging from tweets to professional journals.

The numbers of associated societal publications were not equally distributed over the included scientific publications; one scientific author of multiple societal publications could have skewed our results. We therefore recommend further research on the roles of individual researchers in writing responsible societal publications.

Our sample of scientific publications was small and insufficiently wide-ranging to determine the prevalence of reporting inadequacies and inconsistencies across the field of HSR internationally.

## Interpretation

Our results indicate that most societal publications contain inconsistencies or replicated reporting inadequacies. Inconsistencies are not necessarily negative, as they may correct an inadequacy in the scientific publication. Moreover, reporting inadequacies we identified in this study were not necessarily 'bad'. There is no straightforward rule for what is allowed in terms of rhetorical wordings or simplifications of scientific results in either scientific or societal publications. However, the current discussion on public reporting is focused too narrowly on exaggeration and causality [10, 11]. There is little debate on questions such as whether conclusions and recommendations are adequately reported in scientific literature, the extent to which messages in societal publications may justifiably be simplified, how much detail needs to be provided, and whether a researcher or journalist may add interpretations in societal publications that would not be accepted in scientific literature.

Reporting inadequacies in scientific publications are commonly replicated in societal publications. Most frequently this involves inadequately reported conclusions, policy and practice recommendations, and titles. It is therefore insufficient to focus merely on preventing inconsistencies in societal publications. We recommend that future studies that assess quality in societal publications should extend their research questions to analyse this interplay between the reporting in scientific publications and societal publications.

No substantial differences emerged overall between societal publications produced by research institutes or funders and ones written by outsiders. Moreover, none of the included societal publications were written by other secondary authors of a scientific publication. A stronger relation between the involvement of researchers in writing societal publications and consistency with their scientific publications may be desirable. Media pressures, relationships with funders, and journal demands may cause researchers to consciously or unconsciously introduce reporting inadequacies into a scientific publication [29–31].

## Implications and recommendations for policy and practice

The current COVID-19 pandemic shows the impact of disinformation and misinformation (e.g. on trust in government and their measures). Researchers, research institutes and journalists should be attentive to the effects that the rewriting of research results and conclusions in societal publications might have on policy and practice. Additionally, researchers should be aware that reporting inadequacies in their scientific publications may get replicated in societal publications and subsequently affect policy and practice. Routines such as peer feedback in the final stages of publication could prevent such reporting inadequacies from occurring in scientific publications. Such peer feedback is equally relevant to apply to societal publications. Further training and time dedicated to societal reporting and to communicating about scientific work in lay language would better equip researchers to take active roles in the writing of societal publications.

## Conclusion

To improve societal publications on health services research, we should examine not only how consistency with scientific research publications can be achieved, but also how to prevent scientific reporting inadequacies from being replicated in societal publications. HSR institutions, funders, and scientific and societal publication platforms should invest in a supportive

publication culture in order to further incentivise the responsible and skilled involvement of researchers in writing both scientific and societal publications.

## Supporting information

**S1 Appendix.**
(DOCX)

**S2 Appendix.**
(DOCX)

## Acknowledgments

We thank the Dutch HSR institutions that participated in this study: Erasmus MC, Department of Public Health; Erasmus University, Erasmus School of Health Policy and Management; Leiden University Medical Centre, Departments of Medical Decision Making and Public Health and Primary Care; Maastricht University, Health Services Research; Netherlands Institute for Health Services Research (NIVEL); Radboud UMC, IQ Healthcare; National Institute for Public Health and the Environment (RIVM); University of Groningen, Faculty of Economics and Business; Tilburg University, Social and Behavioural Sciences, Tranzo; Trimbos Institute; University Medical Centre Utrecht, Julius Centre for Health Sciences and Primary Care; and Amsterdam UMC, locations VU Medical Centre and Academic Medical Centre.

## Author Contributions

**Conceptualization:** Reinie G. Gerrits, Michael J. van den Berg, Anton E. Kunst, Niek S. Klazinga, Dionne S. Kringos.

**Data curation:** Reinie G. Gerrits.

**Formal analysis:** Reinie G. Gerrits, Dionne S. Kringos.

**Funding acquisition:** Reinie G. Gerrits, Michael J. van den Berg, Dionne S. Kringos.

**Investigation:** Reinie G. Gerrits.

**Methodology:** Reinie G. Gerrits, Michael J. van den Berg, Anton E. Kunst, Niek S. Klazinga, Dionne S. Kringos.

**Project administration:** Reinie G. Gerrits, Niek S. Klazinga, Dionne S. Kringos.

**Resources:** Reinie G. Gerrits.

**Supervision:** Michael J. van den Berg, Anton E. Kunst, Niek S. Klazinga, Dionne S. Kringos.

**Validation:** Michael J. van den Berg, Anton E. Kunst, Niek S. Klazinga, Dionne S. Kringos.

**Writing – original draft:** Reinie G. Gerrits.

**Writing – review & editing:** Reinie G. Gerrits, Michael J. van den Berg, Anton E. Kunst, Niek S. Klazinga, Dionne S. Kringos.

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
