## [Decision Letter · Decision Letter 0]

4 Sep 2020

PONE-D-20-23277

Reporting Health Services Research to a broader public: An exploration of inconsistencies and reporting inadequacies in societal publications

PLOS ONE

Dear Dr. Kringos,

Thank you for submitting your manuscript to PLOS ONE. After careful consideration, we feel that it has merit but does not fully meet PLOS ONE’s publication criteria as it currently stands. Therefore, we invite you to submit a revised version of the manuscript that addresses the points raised during the review process.

The reviewers noted the importance and novelty of the topic. However, to further assess the manuscript, they requested much more comprehensive reporting of the Methods in place of referencing a past publication. Particularly, they perceived major inconsistencies between the purported study design (a qualitative content analysis) and the analysis strategy and presentation of results (which largely included statistics). You will note that Reviewers 1 and 3 offer conflicting suggestions as to how to remedy the incongruence between the design, methods, analysis, and reporting, however, they were in agreement that the incongruence needs addressing. The authorship team thus has some important methodological and analytic choices to make and I look forward to receiving a revised version of the manuscript.

We look forward to receiving your revised manuscript.

Kind regards,

Quinn Grundy, PhD, RN

Academic Editor

PLOS ONE

Journal Requirements:

Reviewers' comments:

Reviewer's Responses to Questions

**Comments to the Author**

1. Is the manuscript technically sound, and do the data support the conclusions?

Reviewer #1: Partly

Reviewer #2: Yes

Reviewer #3: Partly

2. Has the statistical analysis been performed appropriately and rigorously? 

Reviewer #1: No

Reviewer #2: Yes

Reviewer #3: No

3. Have the authors made all data underlying the findings in their manuscript fully available?

Reviewer #1: No

Reviewer #2: No

Reviewer #3: Yes

4. Is the manuscript presented in an intelligible fashion and written in standard English?

Reviewer #1: Yes

Reviewer #2: Yes

Reviewer #3: Yes

5. Review Comments to the Author

Reviewer #1: The authors sought to investigate possible inconsistencies and reporting inadequacies in non-scientific societal publications based on published health services research. This is an important topic of increasing interest in both academic, healthcare and public spheres. The rationale is well-written and clear. I have a number of questions and concerns, that if addressed will strengthen the paper.

Major

1. There is confusion as to the design of this study. It is described as a qualitative study, yet a number of statistical results are reported, which is particularly prominent in the abstract. The statistical results are of little use, especially given questions as to the inclusion criteria related to the studies. I suggest deleting the statistical analysis altogether, as well as the emphasis on the number of publications that authors deemed as engaging in specific inaccuracies. The number of publications with each type of identified inaccuracy are not the purpose of the study. I would recommend authors focus on the types of inaccuracies found and describing those. The descriptions are really limited.

2. The description of a result as a “statistical trend” is problematic and should be deleted (lines 318-319). Wood et al. Trap of trends to statistical significance: likelihood of near significant P value becoming more significant with extra data. BMJ 2014; 348 doi: https://doi.org/10.1136/bmj.g2215

3. More information on the inclusion criteria of scientific publications is required for this paper to stand alone. Are qualitative and quantitative studies included? Why were 46 publications included (23 for each group of reporting adequacies)? Later this is unclear on pg 10, lines 203-205. A flow diagram may be helpful here. Later in the discussion, authors mention that qualitative and quantitative studies were included.

4. It’s not clear if societal publications were limited to Dutch or English, or both? Same for scientific publications. Please clarify. If there were different languages used in societal vs. scientific publications, this needs to be explained and examined in greater detail.

5. It is not clear why societal publications that only included results were excluded?

6. It is unclear why the dataset cannot be shared if all documents are publicly available.

7. I suggest providing quotes to provide examples of types of inconsistencies in reporting.

8. In the discussion authors could provide a more fulsome discussion of the limitations present in dissemination by societal publications. There is pressure from media to utilize sensational headlines, to focus on recommendations, and to communicate findings with extremely limited text space, etc.

Reviewer #2: This is an interesting paper, which addresses an important topic. The paper is clear and well written. I have highlighted a few areas that requires further consideration/clarification.

Authors have focused on the role of first scientific author in inconsistencies appearing in societal publications. However, I wonder if they have also looked at the involvement of other authors. Any of the authors of the scientific publications (rather than first author) may author some societal publications. As it is, it seems some of these may have been classified as “no involvement of scientific author”. Please clarify. If there are any possibilities of misclassification, it should be discussed in the limitations.

The authors included societal publications up to two years before the scientific publication. Some of the earlier societal publications may be based on interim results. Potential implications of this should also be discussed in the discussion, especially if significant proportion of the societal publications were published about two years before the scientific publication. In addition, Table 1 should show the distribution of the societal publications based on time between the scientific publication and the societal publication. For example, the table can show how many societal publications were published two years before the scientific publication, one year before scientific publication, same year as scientific publication and then one year after the scientific publication.

Some projects may involve various work packages, which may be reported in different scientific publications? Therefore, a societal publication may report a different but related work package to the scientific publication. It is not clear if the authors considered such issues. This may account for some of the inconsistencies, such as new results being introduced that were not reported in the corresponding scientific publication. Please clarify.

Reporting inadequacies were categorised into high vs low but it may be good to also have a further insight to the distribution of inadequacies. For example, a range (of the number of inconsistencies reported for the 43 publications) would be useful. Are there any scientific publications with no reporting inadequacies in any of the associating societal publications? What was the maximum number of inconsistencies identified for one publication?

I would also suggest that the authors report the distribution of reporting inadequacies in a table/figure showing the 35 possible inadequacies and the corresponding number of publications. This is similar to what was presented in Figure 2 of their previous publication (reference 21) but should be based on the sample (43 scientific publications) used in this study. Such aggregated data of the distribution of reporting inadequacies would not compromise anonymity. I believe it would be useful for the readers.

Are the authors able to clarify how the sample size of 23 in each group was determined?

In some subheadings (such as line 224, 236, 246 etc), it is not clear why the figures (such as, “64 societal publications, 41.0%”) were presented in the sub-heading rather than the main body of the sections.

Reviewer #3: This manuscript aims to compare articles and their respective media reports, whenever available. Below see my comments. I suggest a major throughout the points I have stressed. I congratulate the authors for addressing this relevant topic and please see my comments as a way to improve your research.

-----

33 “such translated informations”

33 When you introduce you will investigate inconsistencies, I was expecting X between Y. Thus, please re-organize in a way you may allow readers to understand who is the comparator immediately.

37 Please describe in the methods how you did the content analysis and translated it to numerical variables for the chi-squared test. This journal does not limit the word count in the abstract so please be as detailed as you can be.

37 How much of the 43 and the 156 pieces, respectively, did you extract variables? How much missing variables did occur? How many pieces were unable to be extracted in total?

37 The methods seem completely incomplete in the abstract.

45 About the results – and the same is true for the full text: I feel completely lost because of the absence of proportions. Chi-square statistics won’t tell the readers the full info neither the p-values. Please insert counts and proportions with confidence intervals. Suggest: a table in the full text and in the wrote text, only counts/proportions with confidence intervals.

49 Would you have a stronger suggestion for them? How about peer reviewing of societal publications? Do you think illiteracy of readers is the confounder, rather than discrepancies of scientific and societal publications?

60 Please remove “scientific research”

65 Do you really think that policymakers use cross-media to support decisions?

67 This is positive, not negative.

87 Messages and conclusions in scientific publications are too much poor reported. I suggest stressing the phrase and include references (plenty of David Moher, An Wen Chan etc).

103 I suggest collating aims of study in the last paragraph of the Introduction

120 As a spin-off of a paper, I suggest expanding the methods. This habit of to cite “the methods are described elsewhere” is inadequate nowadays and if you insist on this, you are being inconsistent with your own study, that is about reporting.

126 Again

131 From the beginning of the paragraph: it seems to me the sample was chose by a counting until a sufficient number for analysis was achieved. Is that correct?

134 What you considered as inadequacies?

155 How did you ensure societal publications and scientific pieces were matching?

173 Could you provide to the readers the coding?

193 I am really really interest in seeing the risk ratio of the association you did infer. The chi-square test can provide it (and odds ratios, or any ratio in which a contingence can permit) if you set the proper coding for it. In your study. You only showed to the readers if is there association or not, but not the magnitude of it. Thus, here is my recommendation (PS: do not forget of counts and confidence intervals also once relative measures could spin results).

207 Methods are not important to persuade people? Do you, for example, you could be persuaded only by the methods of a given research in a newspaper? I do not agree with such exclusions.

226 How did you consider “partially “?

236 This item alerts me. What is the probability of selective outcome reporting of one of the parts? An Wen Chan has an extensive work about this and Evan Mayo-Wilson about the consistencies of results. Even though I am comparing apples and oranges, maybe an exploratory investigation of such original studies and their registries could be welcome, or an association of publication bias, source of funding and industry, which could reflect in the societal publication. It is even more clear to investigate when you cite that only one negative result (called by non-significant) was found. It goes towards to the literature that deals with scientific pieces.

246 Recommendations by societal pieces are completely useless for me. Thank you for the data.

254 Societal pieces with poorly reported information and spin is also useless for me. Thank you for the data againg.

267 Causality? That’s terrific. Even well conducted RCTs don’t dare to accuse causality. Thank you for the data.

272 Generalizations? The same.

278 A result of what I meant above.

282 Replication of inadequacy: I congratulate you by addressing this very important topic. If you can just clarify for me a point I didn’t catch, I would be more than welcome. You found that barely 50% of your pieces matched in inadequacy by a lot of potential reasons. However, in the description below, the numbers do not reflect the same for a case of interpretation. I know this might be a source of data collection (methods) rather than internal validity – I needed to deal in data set of a study of mine about this situation. Is that the case? If yes, I strongly recommend you inform authors with a glossary of variables and how they were collected. If not, any potential reason for the discrepancies between the data?

307 Societal publications do not necessarily need to involve first authors or any author once the publication is online. However, in my point of view, if the first author (or any author) could be included and work together journalists and persons of communication, the literacy of the general population would improve exponencially by a combination of expertises. So, nice to see you X2. I won’t say you have a negative result – maybe there is an erratic interpretation (in advance, please remove the term trend in all citations) of what a p-value is. A p-value basically tells you what the probability of the data of given distribution is is embedded in the other distribution, and you set the tolerable limit for your inference in terms of dispersion. Do you really think that 2.0% of an increase of a bit of noise in your distribution would blunt your null hypothesis test? Please re-write it accordingly to the American Statistical Association. The same is for any other inferential test in your study. Finally, about stats, don’t claim for efficacy/association based on the p-value. Please interpret the X2 statistics (or the relative risk or any association measure I recommended before).

378 Your results have more than “some” (what is not bad). Just see the numbers. Please re-phrase it accordingly.

407 You can’t say it affects interpretation of general population. You didn’t measure it. As a legacy of COVID-19, I think one of the most important things we as scientists need to catch is how large is the noise in the literature and the attempt to publish whatever you have in hands. Given this, combined with the illiteracy of the population about evidence-based decisions, science, treatments, etc – which will be another legacy of COVID-19, I definitely would conclude this paper you are writing based in what is happening with evidence and media (no politics, please). This could be an important, reasonable recommendation.

6. PLOS authors have the option to publish the peer review history of their article (what does this mean?). If published, this will include your full peer review and any attached files.

Reviewer #1: No

Reviewer #2: No

Reviewer #3: **Yes: **Lucas Helal

---

## [Author Response · Author response to Decision Letter 0]

4 Dec 2020

Revisions “Reporting Health Services Research to a broader public"

Dear editor and reviewers, 

Thank you for the opportunity to revise our manuscript “Reporting Health Services Research to a broader public”

We thank the reviewers for their valuable feedback. We describe how we have addressed the comments below. 

Sincerely, 

R.G. Gerrits

M.J. van den Berg

A.E. Kunst

N.S. Klazinga

D.S. Kringos

Reviewer #1: The authors sought to investigate possible inconsistencies and reporting inadequacies in non-scientific societal publications based on published health services research. This is an important topic of increasing interest in both academic, healthcare and public spheres. The rationale is well-written and clear. I have a number of questions and concerns, that if addressed will strengthen the paper.

Major

1. There is confusion as to the design of this study. It is described as a qualitative study, yet a number of statistical results are reported, which is particularly prominent in the abstract. The statistical results are of little use, especially given questions as to the inclusion criteria related to the studies. I suggest deleting the statistical analysis altogether, as well as the emphasis on the number of publications that authors deemed as engaging in specific inaccuracies. The number of publications with each type of identified inaccuracy are not the purpose of the study. I would recommend authors focus on the types of inaccuracies found and describing those. The descriptions are really limited.

Reply: Thank you for your comments regarding the design of the study. We understand we have created some confusion with regard to the qualitative and quantitative aims of the study. Our research questions are best answered through the combination of initial qualitative assessment followed by a quantitative description, as is not uncommon for directed content analyses (e.g. Hsieh H-F, Shannon SE.(200): doi:10.1177/1049732305276687). The numerical insights are important to answer in particular research question three, and further provide insight in what consistencies and reporting inadequacies are replicated, and which seem to occur the most. We have adapted our description and wording in the methods section to better characterize the mixed qualitative/quantitative nature of the study whilst emphasizing the description of types and relative importance of inaccuracies found. 

2. The description of a result as a “statistical trend” is problematic and should be deleted (lines 318-319). Wood et al. Trap of trends to statistical significance: likelihood of near significant P value becoming more significant with extra data. BMJ 2014; 348 doi: https://doi.org/10.1136/bmj.g2215

Reply: We have deleted the mention of statistical trends in our paper. 

3. More information on the inclusion criteria of scientific publications is required for this paper to stand alone. Are qualitative and quantitative studies included? Why were 46 publications included (23 for each group of reporting adequacies)? Later this is unclear on pg 10, lines 203-205. A flow diagram may be helpful here. Later in the discussion, authors mention that qualitative and quantitative studies were included.

Reply: Both quantitative and qualitative studies are included. We purposely included 46 publications to gain an even distribution of QRPs in the original scientific publications. We aimed to include publications with the most and the least QRPs,so we might detect differences in the replication reporting inadequacies. We have clarified this by adding this description under the heading subsample of scientific publications. Moreover, We have added a flow diagram to show the selection process as appendix A. 

4. It’s not clear if societal publications were limited to Dutch or English, or both? Same for scientific publications. Please clarify. If there were different languages used in societal vs. scientific publications, this needs to be explained and examined in greater detail.

Reply: Scientific publications were only published in English language journals. Only Dutch language publications were included as societal publications, aside from twitter where the (English language) title was used as a message. We have clarified this in the manuscript under the heading: Sample of societal publications 

5. It is not clear why societal publications that only included results were excluded?

Reply: Thank you for noticing this mistake in wording. We intended to communicate that no societal publications that were solely describing methodology without reference to the results was included. This was adjusted in the manuscript.

6. It is unclear why the dataset cannot be shared if all documents are publicly available.

Reply: To protect the privacy of Dutch authors/researchers whos work was included in the study we complied with existing privacy rules. Although underlying documents are publicly available, making the data set public would unnessesary expose individuals and undermine the trust with which this study was executed in the Dutch HSR community. This procedure was agreed upon with all partner institutes at the start of the study and formalized in a Data Management procedure, and formed an important basis for transparency and trust to allow for collaboration between (often competing) institutes and authors in this study. 

7. I suggest providing quotes to provide examples of types of inconsistencies in reporting.

Reply: Thank you for this suggestion. Due to protection of the authors, we use no direct quotes as this will make the publication traceable. We have however, added some elaboration where we considered it helpful to further illustrate the types. 

8. In the discussion authors could provide a more fulsome discussion of the limitations present in dissemination by societal publications. There is pressure from media to utilize sensational headlines, to focus on recommendations, and to communicate findings with extremely limited text space, etc.

Reply: There is not a strong evidence-base of such pressure executed by media. We can only address the clearly documented expectation by funders and academia that scientists report to non-scientific audiences in lay-terms their results. 

Reviewer #2: This is an interesting paper, which addresses an important topic. The paper is clear and well written. I have highlighted a few areas that requires further consideration/clarification.

Authors have focused on the role of first scientific author in inconsistencies appearing in societal publications. However, I wonder if they have also looked at the involvement of other authors. Any of the authors of the scientific publications (rather than first author) may author some societal publications. As it is, it seems some of these may have been classified as “no involvement of scientific author”. Please clarify. If there are any possibilities of misclassification, it should be discussed in the limitations.

Reply: The category “no involvement of the scientific author” includes publications where none of the authors were involved. This means that in none of the assessed scientific publications secondary authors were named as writers. We clarified this in the manuscript. 

The authors included societal publications up to two years before the scientific publication. Some of the earlier societal publications may be based on interim results. Potential implications of this should also be discussed in the discussion, especially if significant proportion of the societal publications were published about two years before the scientific publication. In addition, Table 1 should show the distribution of the societal publications based on time between the scientific publication and the societal publication. For example, the table can show how many societal publications were published two years before the scientific publication, one year before scientific publication, same year as scientific publication and then one year after the scientific publication.

Reply: We noted how many publications were published before and after the scientific publication. Only a small sample was published before (19.2%). We included these numbers in the manuscript. Because only 20 percent was published in advance, we therefore don’t consider this will significantly impact the findings. 

Some projects may involve various work packages, which may be reported in different scientific publications? Therefore, a societal publication may report a different but related work package to the scientific publication. It is not clear if the authors considered such issues. This may account for some of the inconsistencies, such as new results being introduced that were not reported in the corresponding scientific publication. Please clarify.

Reply: Separate results from related studies and publications were taken into account. We did not mark those results as inconsistent, but excluded them from our analyses. We solely included results that could only have resulted from the same study that was published. These results were presented in the societal publication to be part of the particular published scientific study. 

Reporting inadequacies were categorized into high vs low but it may be good to also have a further insight to the distribution of inadequacies. For example, a range (of the number of inconsistencies reported for the 43 publications) would be useful. Are there any scientific publications with no reporting inadequacies in any of the associating societal publications? What was the maximum number of inconsistencies identified for one publication?

Reply: Scientific publications defined as low had 0-5 reporting inadequacies identified. Those ranked as high had 7-18 reporting inadequacies. The maximum number of inconsistencies for one publication gives no particular information. This number is dependent on the number and type of societal publications per scientific publication, which varies widely across the sample. Therefore, no comparison can be made.

I would also suggest that the authors report the distribution of reporting inadequacies in a table/figure showing the 35 possible inadequacies and the corresponding number of publications. This is similar to what was presented in Figure 2 of their previous publication (reference 21) but should be based on the sample (43 scientific publications) used in this study. Such aggregated data of the distribution of reporting inadequacies would not compromise anonymity. I believe it would be useful for the readers.

Reply: The scatterplot in our previous publication has the primary function of showing the co-occurrence of QRPs in publications. This was possible because scientific publication is standardized and all publications had the same chance of including the same QRPs. This is not the case for the current study. 

Co-occurrence of replicated reporting inadequacies is dependent on the scientific publication having this QRP. However, not all 43 scientific publications contain the same QRPs. Moreover, societal publications vary widely in content and description. In a short societal publication, the chance for replicating multiple reporting inadequacies is low. Any direct comparison between studies would therefore provide no insight in co-occurrence. Because we believe such a graph may easily be misinterpreted, we decided not to include it in the study. 

Are the authors able to clarify how the sample size of 23 in each group was determined?

Reply: We purposely included 46 publications to gain an even distribution of reporting inadequacies in the original scientific publications. We aimed to include publications the most and the least reporting inadequacies, to allow for detection of differences in the replication reporting inadequacies. This decision was made so we might see the difference in societal publications associated with publications with a high number of inadequacies (>6) and a low number of inadequacies (<6). This has been clarified in the methods section. 

In some subheadings (such as line 224, 236, 246 etc), it is not clear why the figures (such as, “64 societal publications, 41.0%”) were presented in the sub-heading rather than the main body of the sections.

Reply: We showed the main result besides category heading (e.g. Inconsistencies in reporting generalisations (3 societal publications, 1.7%) to promote reading efficiency. It will save readers from looking for the full count in the text below. It also immediately shows that it concerns the overall count, rather than the subcounts presented in the text. 

Reviewer #3: This manuscript aims to compare articles and their respective media reports, whenever available. Below see my comments. I suggest a major throughout the points I have stressed. I congratulate the authors for addressing this relevant topic and please see my comments as a way to improve your research.

-----

33 “such translated informations”

Reply: We adapted this sentence.

33 When you introduce you will investigate inconsistencies, I was expecting X between Y. Thus, please re-organize in a way you may allow readers to understand who is the comparator immediately.

Reply: Thank you for this suggestion. We have clarified this. 

37 Please describe in the methods how you did the content analysis and translated it to numerical variables for the chi-squared test. This journal does not limit the word count in the abstract so please be as detailed as you can be.

Reply: We added a more detailed description of how we translated the content analyses to the methods section in the abstract. 

37 How much of the 43 and the 156 pieces, respectively, did you extract variables? How much missing variables did occur? How many pieces were unable to be extracted in total?

Reply: As the results were derived from a qualitative coding, no missing values occurred. Because of this we think it not necessary to mention missing data in the manuscript. We added more descriptive information regarding the included publications in our manuscript. 

37 The methods seem completely incomplete in the abstract.

Reply: We added more information to the methods section in the abstract to allow for a better understanding of our approach. 

45 About the results – and the same is true for the full text: I feel completely lost because of the absence of proportions. Chi-square statistics won’t tell the readers the full info neither the p-values. Please insert counts and proportions with confidence intervals. Suggest: a table in the full text and in the wrote text, only counts/proportions with confidence intervals.

Reply: We have adjusted the table including proportions and confidence intervals. As the current way of reporting follows APA standards, we have decided to maintain the description to avoid confusion. 

49 Would you have a stronger suggestion for them? How about peer reviewing of societal publications? Do you think illiteracy of readers is the confounder, rather than discrepancies of scientific and societal publications?

Reply: We have added to the discussion section that peer feedback in the final stages of publication, is equally relevant to be applied to societal publications., 

60 Please remove “scientific research”

Reply: This was adjusted accordingly. 

65 Do you really think that policymakers use cross-media to support decisions?

Reply: Yes. Policy makers use summaries of scientific studies and other published advice to support their decisions. Supporting evidence used in parliament and government often refers to newspaper articles, reports and summaries (as was investigated in: Gerrits, R.G., van den Berg, M.J., Klazinga, N.S. et al. Health Res Policy Sys 17, 55 (2019). https://doi.org/10.1186/s12961-019-0461-y) . But hardly ever original academic research publications. In a recent (perhaps anecdotal) example, in the Netherlands the effect of using masks against corona is summarized and communicated through societal publications (occasionally in the form of direct advice). Citing these societal publications, the Dutch government has decided to not make wearing masks mandatory. Abroad however, based on the same evidence, different conclusions are communicated in societal publications, and consequentially, different measures are taken. 

67 This is positive, not negative.

Reply: We did not intent to make this sound negative, but describe a trend neutrally. As the parentheses might have given the impression it was a negative interpretation, we removed them. 

87 Messages and conclusions in scientific publications are too much poor reported. I suggest stressing the phrase and include references (plenty of David Moher, An Wen Chan etc).

Reply: We have stressed the phrase and added references. 

103 I suggest collating aims of study in the last paragraph of the Introduction

Reply: We included a detailed description of the aims to promote readability. The aims differ in a subtle manner, which makes it difficult to collate them without creating confusion. 

120 As a spin-off of a paper, I suggest expanding the methods. This habit of to cite “the methods are described elsewhere” is inadequate nowadays and if you insist on this, you are being inconsistent with your own study, that is about reporting.

Reply: Thank you for this suggestion. We have expanded our description of the assessment of the reporting inadequacies and added a list of inadequacies to the appendix. 

126 Again

Reply: We have described the methods in more detail. 

131 From the beginning of the paragraph: it seems to me the sample was chose by a counting until a sufficient number for analysis was achieved. Is that correct?

Reply: That is correct. We have clarified this. 

134 What you considered as inadequacies?

Reply: We have added the list of reporting inadequacies to Appendix B. This will provide a complete description to the readers. A sentence to explain this was added.

155 How did you ensure societal publications and scientific pieces were matching?

Reply: This was done based on content. Either the authors, scientific publication, research institute and or research programmed were mentioned. Further, the descriptions and the methods of the study show whether the societal and scientific publication described the same study. Any societal publication that seemed to refer to another similar or related study was not included. However, we did not identify any societal publications where the relation was unclear. A sentence to explain this was added under the heading of Sample of societal publications.

173 Could you provide to the readers the coding?

Reply: The coding scheme has been added as Appendix C. 

193 I am really really interest in seeing the risk ratio of the association you did infer. The chi-square test can provide it (and odds ratios, or any ratio in which a contingence can permit) if you set the proper coding for it. In your study. You only showed to the readers if is there association or not, but not the magnitude of it. Thus, here is my recommendation (PS: do not forget of counts and confidence intervals also once relative measures could spin results).

Reply: We added the Odds Ratio to table 2, which indeed is easier to interpret for the reader. 

207 Methods are not important to persuade people? Do you, for example, you could be persuaded only by the methods of a given research in a newspaper? I do not agree with such exclusions.

Reply: While methodology is an important tool to aid persuasion, our study was not focused on the description of methodology. The excluded societal publications contained only a mention that the study was being performed and a description through which methods. This was not within the scope of this study.

226 How did you consider “partially “?

Reply: In agreement with the advice provided by reviewer 1, we added further explanation to illustrate this. 

236 This item alerts me. What is the probability of selective outcome reporting of one of the parts? An Wen Chan has an extensive work about this and Evan Mayo-Wilson about the consistencies of results. Even though I am comparing apples and oranges, maybe an exploratory investigation of such original studies and their registries could be welcome, or an association of publication bias, source of funding and industry, which could reflect in the societal publication. It is even more clear to investigate when you cite that only one negative result (called by non-significant) was found. It goes towards to the literature that deals with scientific pieces.

Reply: we will take your suggestion into account for further research to explore this topic more in depth, which goes beyond the current scope of this explorative work for the field of health services research.

282 Replication of inadequacy: I congratulate you by addressing this very important topic. If you can just clarify for me a point I didn’t catch, I would be more than welcome. You found that barely 50% of your pieces matched in inadequacy by a lot of potential reasons. However, in the description below, the numbers do not reflect the same for a case of interpretation. I know this might be a source of data collection (methods) rather than internal validity – I needed to deal in data set of a study of mine about this situation. Is that the case? If yes, I strongly recommend you inform authors with a glossary of variables and how they were collected. If not, any potential reason for the discrepancies between the data?

Reply: The numbers seem to differ because these are respectively the number of identified inadequacies and the number of scientific publications. As multiple inadequacies could be identified in a scientific publication, these figures are slightly different. 

307 Societal publications do not necessarily need to involve first authors or any author once the publication is online. However, in my point of view, if the first author (or any author) could be included and work together journalists and persons of communication, the literacy of the general population would improve exponentially by a combination of expertises. 

Reply: We agree with you.

So, nice to see you X2. I won’t say you have a negative result – maybe there is an erratic interpretation (in advance, please remove the term trend in all citations) of what a p-value is. A p-value basically tells you what the probability of the data of given distribution is is embedded in the other distribution, and you set the tolerable limit for your inference in terms of dispersion. 

Reply: We have removed this term.

Do you really think that 2.0% of an increase of a bit of noise in your distribution would blunt your null hypothesis test? Please re-write it accordingly to the American Statistical Association. The same is for any other inferential test in your study. Finally, about stats, don’t claim for efficacy/association based on the p-value. Please interpret the X2 statistics (or the relative risk or any association measure I recommended before).

Reply: We replaced the chi-square with odds ratios, which makes the associations easier to interpret. Indeed, we don’t draw bold conclusions based on this analyses and associations, which are relatively weak. 

378 Your results have more than “some” (what is not bad). Just see the numbers. Please re-phrase it accordingly.

Reply: We removed the word ‘some’.

407 You can’t say it affects interpretation of general population. You didn’t measure it. As a legacy of COVID-19, I think one of the most important things we as scientists need to catch is how large is the noise in the literature and the attempt to publish whatever you have in hands. Given this, combined with the illiteracy of the population about evidence-based decisions, science, treatments, etc – which will be another legacy of COVID-19, I definitely would conclude this paper you are writing based in what is happening with evidence and media (no politics, please). This could be an important, reasonable recommendation.

Reply: We contextualized the paper in the current context in the discussion by mentioning that the current COVID-19 pandemic has showed the impact of disinformation and misinformation.

---

## [Decision Letter · Decision Letter 1]

18 Jan 2021

PONE-D-20-23277R1

Reporting Health Services Research to a broader public: An exploration of inconsistencies and reporting inadequacies in societal publications

PLOS ONE

Dear Dr. Kringos,

Thank you for submitting your manuscript to PLOS ONE. After careful consideration, we feel that it has merit but does not fully meet PLOS ONE’s publication criteria as it currently stands. Therefore, we invite you to submit a revised version of the manuscript that addresses the points raised during the review process.

The reviewers again had the opportunity review the manuscript and found the reporting to be greatly strengthened. The introduction and the manuscript are reading really well and are much improved. However, there are a few reporting considerations, a couple points of clarification and revision of the abstract that need to happen before we can reach a final decision. I ask that you address the following minor comments:

Please note in your limitations section that your analysis including odds ratios does not adjust for confounders, so please note that this is a gross analysis.  

Many of the changes you made in the main text to improve the clarity of reporting have not been reproduced in the Abstract. Please address the following in the abstract:

- define societal publications in the first sentence as “such as press releases, newspapers, social media, internet postings or professional journals.”

- describe as a content analysis in a way that reflects the Methods in the main text

- a couple of typo  “inconsistencies” instead of “inconstancies” also typo with “all” at the end of the line line 42

- reproduce the language from the main text in describing the coding process

- Line 44- replace “After all documents were coded, counts per code were calculated,” with suggested wording, “Descriptive frequencies were calculated for all variables of interest.”

- please round percentages to one decimal place

- First two lines of the abstract results are confusing; please just state how many scientific and societal publications were included.

- Please also rephrase this sentence lines 54-55: “Reporting inadequacies in 51.2% of the scientific publications were replicated in associated societal publications (28.9%).” I don’t understand what the proportions refer to.

In the main text, please address the following:

- Introduction, line 100, typo “poorly” reported

- Please spell out “questionable research practices” throughout the manuscript as it is a non-standard acronym 

- What is a “directed” qualitative content analysis? You provide a citation later in the paper, but if this is a particular method, please cite here as well. Also, please define what you mean by “directed.” I think simply stating that this study design is a “content analysis” will address some confusing – content analyses often have both qualitative and quantitative analyses. Please also reflect whatever changes you make in the abstract.

- Please make Appendix B a Figure to be included in the manuscript. This figure should also be referenced not in the Methods, but the first paragraph of the Results. I am still rather confused about the 43 initially included scientific HSR publications and then the 46 included scientific HSR publications and many of the numbers reported in the Methods section, but they are much clearer in the first paragraph of the Results.  It will likely add clarity if you simply describe the search and screening strategy in the Methods in general terms, but leave the details about the specific numbers to this paragraph in the Results. Please also reflect any changes to the main text in the abstract.

- I would suggest round proportions to whole numbers throughout the manuscript.

- Wherever you report a proportion, please report also the accompanying (numerator/denominator). Similarly, where you report numerators (e.g. top of page 13, lines 259-272), report the corresponding denominators and proportions (or, simply present the findings qualitatively as I’m not sure these counts really add much). See, for examples, the section beginning line 357 “Role of the first scientific author."

- The sub-headings beginning page 12 (e.g. “Inconsistencies in conclusions) are helpful. However, I would suggest moving the quantification into a sentence in the paragraph following the sub-headings (e.g. 64/X societal publications, 41%).

- Line 332, when you saw that reporting inadequacies were “replicated” or “reproduced” in the societal publication, do you mean that it was copied verbatim? Or that it was also present in some form? The counts in this section might better reflect the scientific/societal publication *pairs*. Currently, you only describe the proportion of “scientific publications” that had an inadequacy, which is confusing given that you just presented these frequencies. Could you rephrase to describe the matched pairs as the unit of analysis?

We look forward to receiving your revised manuscript.

Kind regards,

Quinn Grundy, PhD, RN

Academic Editor

PLOS ONE

Reviewers' comments:

Reviewer's Responses to Questions

**Comments to the Author**

1. If the authors have adequately addressed your comments raised in a previous round of review and you feel that this manuscript is now acceptable for publication, you may indicate that here to bypass the “Comments to the Author” section, enter your conflict of interest statement in the “Confidential to Editor” section, and submit your "Accept" recommendation.

Reviewer #2: All comments have been addressed

Reviewer #3: All comments have been addressed

2. Is the manuscript technically sound, and do the data support the conclusions?

Reviewer #2: Yes

Reviewer #3: Partly

3. Has the statistical analysis been performed appropriately and rigorously? 

Reviewer #2: Yes

Reviewer #3: Yes

4. Have the authors made all data underlying the findings in their manuscript fully available?

Reviewer #2: Yes

Reviewer #3: No

5. Is the manuscript presented in an intelligible fashion and written in standard English?

Reviewer #2: Yes

Reviewer #3: Yes

6. Review Comments to the Author

Reviewer #2: (No Response)

Reviewer #3: (No Response)

7. PLOS authors have the option to publish the peer review history of their article (what does this mean?). If published, this will include your full peer review and any attached files.

Reviewer #2: No

Reviewer #3: **Yes: **Lucas Helal

---

## [Author Response · Author response to Decision Letter 1]

3 Mar 2021

Revisions “Reporting Health Services Research to a broader public"

Dear editor and reviewers, 

Thank you for the opportunity to make further minor revisions to our manuscript “Reporting Health Services Research to a broader public”.

Below, we explain how we have addressed the comments point-by-point. 

Sincerely, 

R.G. Gerrits

M.J. van den Berg

A.E. Kunst

N.S. Klazinga

D.S. Kringos

Comments regarding Ethics:

Your ethics statement should only appear in the Methods section of your manuscript.

Reply: we have moved our ethics statement to the first paragraph of the methods section.

Comments regarding the abstract:

- Please note in your limitations section that your analysis including odds ratios does not adjust for confounders, so please note that this is a gross analysis. 

- Many of the changes you made in the main text to improve the clarity of reporting have not been reproduced in the Abstract. Please address the following in the abstract:

- define societal publications in the first sentence as “such as press releases, newspapers, social media, internet postings or professional journals.”

Reply: we added these suggestions to the abstract, keeping in mind the limit of 300 words. 

Comment: describe as a content analysis in a way that reflects the Methods in the main text

Reply: Given the allowed limit of 300 words, we have expanded the description of the methods section, but kept it focused on its essential elements.

Comment: a couple of typo “inconsistencies” instead of “inconstancies” also typo with “all” at the end of the line line 42

Reply: We have closely re-read the text for consistent use of the word “inconsistencies” and corrected typos.

Comment: reproduce the language from the main text in describing the coding process.

Reply: Given the allowed limit of 300 words, we have expanded the description of the methods section, but kept it focused on its essential elements, which does not allow room for such detailed steps. 

Comment: Line 44- replace “After all documents were coded, counts per code were calculated,” with suggested wording, “Descriptive frequencies were calculated for all variables of interest.”

Reply: we adapted this.

Comment: please round percentages to one decimal place

Reply: currently all our percentages are rounded to one decimal place, however, percentages in the introduction are rounded as they were presented as such in the referenced literature. 

Comment: First two lines of the abstract results are confusing; please just state how many scientific and societal publications were included.

Reply: These two sentences were replaced with a simpler description. 

Comment: Please also rephrase this sentence lines 54-55: “Reporting inadequacies in 51.2% of the scientific publications were replicated in associated societal publications (28.9%).” I don’t understand what the proportions refer to.

Reply: We rephrased this sentence. 

Comments: In the main text, please address the following:

- Introduction, line 100, typo “poorly” reported

- Please spell out “questionable research practices” throughout the manuscript as it is a non-standard acronym 

- What is a “directed” qualitative content analysis? You provide a citation later in the paper, but if this is a particular method, please cite here as well. Also, please define what you mean by “directed.” I think simply stating that this study design is a “content analysis” will address some confusing – content analyses often have both qualitative and quantitative analyses. Please also reflect whatever changes you make in the abstract.

Reply: We adapted the text in accordance with these suggestions and added a sentence explaining the core concept of a directed content analyses approach. 

Comment: Please make Appendix B a Figure to be included in the manuscript. This figure should also be referenced not in the Methods, but the first paragraph of the Results.

Reply: The flow chart was entered as a figure to the results section. 

 I am still rather confused about the 43 initially included scientific HSR publications and then the 46 included scientific HSR publications and many of the numbers reported in the Methods section, but they are much clearer in the first paragraph of the Results. It will likely add clarity if you simply describe the search and screening strategy in the Methods in general terms, but leave the details about the specific numbers to this paragraph in the Results. Please also reflect any changes to the main text in the abstract.

Reply: All mentioning of numbers of societal publications was left out of the method section, limiting reporting of the selection results to the results section. 

Comment: I would suggest round proportions to whole numbers throughout the manuscript.

Reply: In line with the earlier suggestion, we decided to maintain a decimal point reporting. 

Comment: Wherever you report a proportion, please report also the accompanying 

(numerator/denominator). Similarly, where you report numerators (e.g. top of page 13, lines 259-272), report the corresponding denominators and proportions (or, simply present the findings qualitatively as I’m not sure these counts really add much). See, for examples, the section beginning line 357 “Role of the first scientific author."

Reply: All percentages are now preceded by numerator/denominator. 

Comment: The sub-headings beginning page 12 (e.g. “Inconsistencies in conclusions) are helpful. However, I would suggest moving the quantification into a sentence in the paragraph following the sub-headings (e.g. 64/X societal publications, 41%).

Reply: We moved the numbers into the suggested paragraph. 

Comment: Line 332, when you saw that reporting inadequacies were “replicated” or “reproduced” in the societal publication, do you mean that it was copied verbatim?

Reply: As stated in the methods “A message that identically reproduced the reporting inadequacy was marked as a ‘replicated reporting inadequacy’. The scientific publication is in English and the societal publications were Dutch. Thus, an identical reproduction refers to a direct translation. 

Comment: Or that it was also present in some form? The counts in this section might better reflect the scientific/societal publication pairs. Currently, you only describe the proportion of “scientific publications” that had an inadequacy, which is confusing given that you just presented these frequencies. Could you rephrase to describe the matched pairs as the unit of analysis?

Reply: For clarification we added the number of societal publications matching with the scientific publications.

---

## [Editor Report · Decision Letter 2]

5 Mar 2021

Reporting Health Services Research to a broader public: An exploration of inconsistencies and reporting inadequacies in societal publications

PONE-D-20-23277R2

Dear Dr. Kringos,

We’re pleased to inform you that your manuscript has been judged scientifically suitable for publication and will be formally accepted for publication once it meets all outstanding technical requirements.

Kind regards,

Quinn Grundy, PhD, RN

Academic Editor

PLOS ONE
---

## [Editor Report · Acceptance letter]

16 Mar 2021

PONE-D-20-23277R2 

Reporting Health Services Research to a broader public: An exploration of inconsistencies and reporting inadequacies in societal publications 

Dear Dr. Kringos:

I'm pleased to inform you that your manuscript has been deemed suitable for publication in PLOS ONE. Congratulations! Your manuscript is now with our production department. 

Kind regards, 

on behalf of

Dr. Quinn Grundy 

Academic Editor

PLOS ONE